# IMPROVING CONFIDENT-CLASSIFIERS FOR OUT-OF-DISTRIBUTION DETECTION

## ABSTRACT

Discriminatively trained neural classifiers can be trusted, only when the input data comes from the training distribution (in-distribution). Therefore, detecting out-of-distribution (OOD) samples is very important to avoid classification errors. In the context of OOD detection for image classification, one of the recent approaches proposes training a classifier called "confident-classifier" by minimizing the standard cross-entropy loss on in-distribution samples and minimizing the KL divergence between the predictive distribution of OOD samples in the low-density "boundary" of in-distribution and the uniform distribution (maximizing the entropy of the outputs). Thus, the samples could be detected as OOD if they have low confidence or high entropy. In this paper, we analyze this setting both theoretically and experimentally. We also propose a novel algorithm to generate the "boundary" OOD samples to train a classifier with an explicit "reject" class for OOD samples. We compare our approach against several recent classifier-based OOD detectors including the confident-classifiers on MNIST and Fashion-MNIST datasets. Overall the proposed approach consistently performs better than others across most of the experiments.

## 1 INTRODUCTION

Discriminatively trained deep neural networks have achieved state of the art results in many classification tasks such as speech recognition, image classification, and object detection. This has resulted in deployment of these models in real life applications where safety is paramount (e.g., autonomous driving). However, recent progress has shown that deep neural network (DNN) classifiers make overconfident predictions even when the input does not belong to any of the known classes (Nguyen et al. (2015)). This follows from the design of DNN classifiers that are optimized over in-distribution data without the knowledge of OOD data. The resulting decision boundaries are typically "unbounded/open" as shown in Figure 1a resulting in over-generalization (Spigler (2019), Scheirer et al. (2012)).

There have been many approaches proposed to address this problem under the umbrella of OOD detection[1]. Lee et al. (2018a) propose to explicitly train a classifier using the OOD samples generated by a GAN (Goodfellow et al. (2014a)). They empirically try to show that, for effective OOD detection, the generated OOD samples should follow and be close to the low-density boundaries of in-distribution, and the proposed GAN training indeed tries to do that. A multi-class softmax DNN classifier is trained with in-distribution samples to minimize the standard cross-entropy loss (minimizing the output entropy) and the generated OOD samples are trained to minimize a KL loss that forces the classifier's predictive distribution to follow a uniform one (maximizing the output entropy). The resulting classifier is called a "confident-classifier". One can then classify a sample as being in or out-of distribution based on the maximum prediction probability or the entropy of the output. Sricharan & Srivastava (2018) also follow a similar approach with slight modifications. **Contribution.** One of the key assumptions in Lee et al. (2018a) and Sricharan & Srivastava (2018) is that the effect of maximizing the entropy for OOD samples close to the low-density boundaries of in-distribution might also propagate to samples that are far away from in-distribution. This training is expected to result in "bounded/closed" regions in input space with lower entropy over the in-distribution, and the rest of the region (corresponding to OOD), with higher entropy. The ideal

---

[1]Related work section is in the appendix due to space constraints.

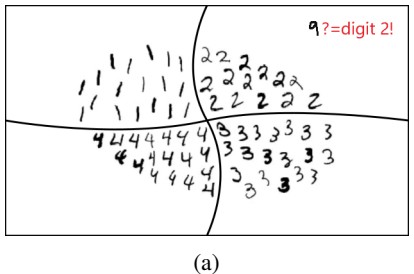 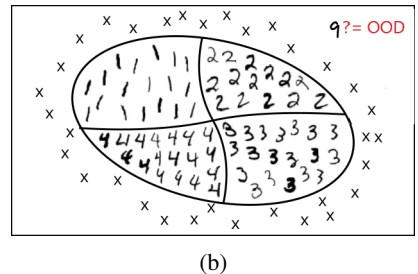

|   (a)   |   (b)   |

Figure 1: Figure shows how the decision boundaries would change and become more bounded when a typical classifier is trained with an auxiliary ("reject") class containing OOD samples. (a) The unbounded decision boundaries of a typical 4-class classifier. Digit 9 is incorrectly classified as digit 2 with very high confidence. (b) A 5-class classifier trained with OOD samples 'x' that are close to in-distribution and form the fifth ("reject") class, resulting in bounded decision boundaries. Digit 9 is correctly classified as belonging to the "reject" (OOD) class.

decision boundary in such a scenario would be as shown in Figure 1b. We find that even though such a solution exists, the proposed training algorithm is unlikely to reach it. We justify this both theoretically and experimentally for a ReLU network (network with ReLU activation units) that was indeed used in Lee et al. (2018a). Assuming training with OOD samples close to the in-distribution boundary, we find that having an explicit reject class for OOD samples results in a solution close to the one depicted in Figure 1b. Therefore we propose to use such a classifier instead. We give intuitive arguments to justify the proposal. This forms the first key contribution of our paper.

Moreover, with toy experiments (refer to section D in appendix) on low-dimensional synthetic data, we analyze if GAN can indeed produce samples that can follow the low-density boundaries of in-distribution. We find that, even though GAN produces samples close to the low-density boundaries of in-distribution, it is unable to cover the whole boundary, thus resulting in a sub-optimal OOD detector when trained on such samples. We therefore propose a novel algorithm to generate "boundary" OOD samples using a manifold learning network, (e.g., variational auto-encoder (VAE)) and show that the generated samples are diverse and cover the in-distribution boundaries better than the method proposed in Lee et al. (2018a). The resulting classifier trained with those samples improves the OOD detection results. This forms the second key contribution of our paper.

## 2 BACKGROUND

Lee et al. (2018a) propose a joint training of GAN and a classifier based on the following objective:

$$\min_G \max_D \min_\theta \underbrace{\mathbb{E}_{P_{in}(\hat{x},\hat{y})}[-\log P_\theta(y=\hat{y}|\hat{x})]}_{(a)} + \beta \underbrace{\mathbb{E}_{P_G(x)}[\mathrm{KL}(\mathcal{U}(y)||P_\theta(y|x))]}_{(b)}$$
$$+ \underbrace{\mathbb{E}_{P_{in}(x)}[\log D(x)] + \mathbb{E}_{P_G(x)}[\log(1-D(x))]}_{(c)} \tag{1}$$

where (b)+(c) is the modified GAN loss and (a)+(b) is the classifier loss ($\theta$ is the classifier's parameter) called the confidence loss. The difference from the regular GAN objective is the additional KL loss in (1), which when combined with the original loss, forces the generator to generate samples in the low-density boundaries of the in-distribution ($P_{in}(x)$) space. $\beta$ is a hyper-parameter that controls how close the OOD samples are to the in-distribution boundary. For the classifier, the KL loss pushes the OOD samples generated by GAN to produce a uniform distribution at the output, and therefore have higher entropy. This enables one to detect OOD samples based on the entropy or the confidence at the output of the classifier.

## 3 WHY MINIMIZING CONFIDENCE LOSS IS INSUFFICIENT FOR OOD DETECTION

Let $f : \mathbb{R}^d \to \mathbb{R}^K$ be the neural network function that maps input in $\mathbb{R}^d$ to $K$ output classes (input to the softmax layer). Let $f_k : \mathbb{R}^d \to \mathbb{R}$ be the function that maps the input to output for a specific class $k \in \{1, 2, 3...K\}$. For a neural network with affine activations (e.g., ReLU and Leaky ReLU), each $f_k$ is a continuous piece-wise affine function over a finite set of polytopes, $\{Q_1, Q_2, \cdots, Q_M\}$ such that $\mathbb{R}^d = \bigcup_{l=1}^M Q_l$, as described in Croce & Hein (2018). This means that each $f_k$ is affine within each $Q_l$ ($l \in \{1, 2, 3...M\}$). If the input space is $\mathbb{R}^d$, some of these polytopes stretch to infinity (grow without bounds). Let $Q_l^\infty \equiv Q_l$ denote these "infinity polytopes". The choice of the neural network structure and the weights define $f_k$'s. Figure 2a illustrates these polytopes and $f_k$'s for a simple 3-class ReLU classifier, where the input space is $\mathbb{R}$. In this example, there are 4 polytopes in which $Q_1^\infty$ and $Q_4^\infty$ stretch to infinity.

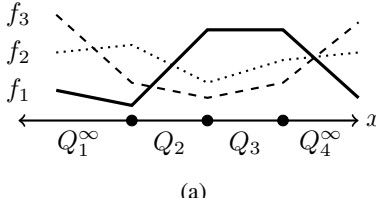
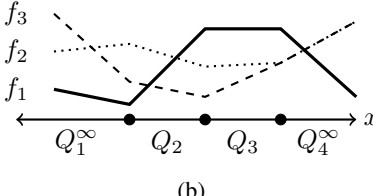

(a)                                               (b)

Figure 2: $f_k$'s and $Q_r$'s for an example 3-class ReLU classifier where the input $x \in \mathbb{R}$. $Q_1^\infty$ and $Q_4^\infty$ are infinity polytopes. (a) For sufficiently large (small) $x$, there is a unique $k^* = 1$ in $Q_4^\infty$ ($k^* = 1$ in $Q_1^\infty$). (b) For sufficiently large $x$, there are multiple $k^*$'s in $Q_4^\infty$ ($k^* = \{2, 3\}$). For sufficiently small $x$, there is a unique $k^* = 3$ in $Q_1^\infty$.

Hein et al. (2019) mathematically show that a ReLU classifier (with softmax output) produces arbitrarily high confidence predictions (approaching 1) far away from the training data in almost all directions on an unbounded input space. This happens over $Q_l^\infty$'s. Their results are summarized as follows.

For any $\boldsymbol{x} \in \mathbb{R}^d$, there exists a $\beta_l > 0$ such that for all $\alpha_l \geq \beta_l$, $\alpha_l \boldsymbol{x} \in Q_l^\infty$. Let $f_k^l(\boldsymbol{x}) = \langle \boldsymbol{v}_k^l, \boldsymbol{x} \rangle + a_k^l$ be the piece-wise affine function for class $k$ over $Q_l$. Let $k^* = \arg\max_k \langle \boldsymbol{v}_k^l, \beta_l \boldsymbol{x} \rangle^2$. Then, as $\alpha_l \to \infty$, the confidence for input $\alpha_l \boldsymbol{x}$ for class $k^*$ becomes arbitrarily high if $k^*$ is unique. i.e,

$$\lim_{\alpha_l \to \infty} \frac{e^{f_{k^*}(\alpha_l \boldsymbol{x})}}{\sum_{l=1}^K e^{f_l(\alpha_l \boldsymbol{x})}} = 1$$

But if there are multiple $k^*$'s, arbitrarily large confidence values cannot be obtained far away from the in-distribution in the direction of $x$. For instance, as shown in Figure 2b[3], for $Q_4^\infty$, $k^* = \{2, 3\}$ and therefore arbitrarily high confidence predictions cannot be achieved as $\alpha_l \to \infty$. Having multiple $k^*$'s for every $Q_l^\infty$ is highly unlikely, given that we are dealing with floating point numbers and also that it is not explicitly enforced during training. Therefore, arbitrarily high confidence values far away from the in-distribution are likely inevitable.

The above analysis is for the case where the input domain is unbounded ($R^d$). For bounded domains (for example, $[0, 1]^d$ for images), as pointed out in Hein et al. (2019), since we cannot let $\alpha_l \to \infty$, the above analysis cannot be directly applied to get arbitrary high confidence values. However the above technique in principle can be applied to increase the prediction confidence for samples far away from the in-distribution. Hein et al. (2019) conduct experiments to support the claim. The theoretical analysis of which can be done as follows. Let $\mathbb{Z}^d$ represent the bounded input domain. Similar to the unbounded case, let $Q_l^\infty$ denote "infinity polytopes" that stretch till the

---

[2]Note, $k^* = \arg\max_k \langle \boldsymbol{v}_k^l, \alpha_l \boldsymbol{x} \rangle = \arg\max_k \langle \boldsymbol{v}_k^l, \beta_l \boldsymbol{x} \rangle, \forall \alpha_l \geq \beta_l$. Also note, we define k* only for infinity polytopes.

[3]Note, for $x \in \mathbb{R}$, $k^* = \arg\max_k [\text{slope of} f_k(\alpha_l x)]$ ($= \arg\max_k [\text{negative slope of} f_k(\alpha_l x)]$) as $\alpha_l \to \infty$ ($\alpha_l \to -\infty$).

bounds of the input domain $\mathbb{Z}^d$. For any $\boldsymbol{x} \in \mathbb{Z}^d$, there exists a $\beta_l > 0$ such that for all $\alpha_l \geq \beta_l$, $\alpha_l \boldsymbol{x} \in Q_l^\infty$. Let $f_k^l(\boldsymbol{x}) = \langle \boldsymbol{v}_k^l, \boldsymbol{x} \rangle + a_k^l$ be the piece-wise affine function for class $k$ over $Q_l$. Let $k^* = \arg\max_k \langle \boldsymbol{v}_k^l, \beta_l \boldsymbol{x} \rangle$. Then, as $\alpha_l$ increases, the confidence for input $\alpha_l \boldsymbol{x}$ for class $k^*$ keeps increasing until the bounds of the domain is reached if $k^*$ is unique. If $f_{k^*}(\beta_l x) >> f_k(\beta_l x)\ \forall k \neq k^*$ (ignoring the effect of bias term for simplicity), the confidence for input $\alpha_l x$ for the class $k^*$ is very high. Therefore, even for the case of bounded input space, one can obtain confidence predictions for OOD samples high enough for it to be considered as in-distribution samples.

**Corollary.** The higher the confidence of the output, the lower is the entropy. Hence a direct corollary of Hein et al. (2019)'s result is that the entropy of the classifier output for data far away from the in-distribution data in all directions would almost always be arbitrarily low (approaching 0) like the in-distribution samples. This makes it almost impossible to detect OOD samples based on the confidence or the entropy of the classifier outputs. For the case of bounded input domain, as one can increase the prediction confidence for OOD samples far from the in-distribution, the entropy of classifier output also decreases making those OOD samples to be classified as in-distribution samples. Therefore, the approaches in Lee et al. (2018a) and Sricharan & Srivastava (2018) would not be applicable.

# 4 ADDING AN EXPLICIT "REJECT" CLASS

When OOD samples are generated close to the in-distribution and follow its low-density boundaries as proposed in Lee et al. (2018a) and Sricharan & Srivastava (2018), we recommend adding an explicit reject class for OOD samples instead of minimizing the loss in Eq.1(b). Let the resulting classifier be called the reject-classifier. By adding an explicit reject class, our goal is to obtain a decision boundary close to the ideal decision boundary shown in Figure.1b, where the decision boundary of a $K + 1$ classifier divides the input space into regions such that the in-distribution region is classified as one of the first $K$ classes and the rest of the region as the $K + 1^{th}$ class, i.e., the reject class. The intuition on how such a decision boundary can be obtained is as follows. The arbitrarily high confidence predictions happen in polytopes that stretch to infinity (or stretch till the bounds of input space in case of bounded input space). Each of the "infinity polytopes" has its own class (or classes), $k^*$(or $k^*$'s) where high confidence predictions occur. If adding an explicit "reject" class results in $k^*$ = reject-class for all the "infinity polytopes" (i.e there is only one $k^*$), the arbitrarily high confidence predictions would only happen at the reject class for OOD samples far-off from training data. Therefore, these samples will be detected as OOD. We argue that in reject-classifier training, since we explicitly maximize the prediction confidence of $K + 1^{th}$-class for boundary OOD samples, we expect the same effect to persist for OOD samples far from the in-distribution as well (i.e., $k^* = K + 1$) resulting in close to ideal decision boundaries depicted in Figure.1b. This claim is supported by our experiments on a toy dataset (Figure. 5) and the superior performance of the reject-classifier over the confident-classifier on MNIST and Fashion MNIST datasets (Table. 1). Note that how close the resulting decision boundary of the reject-classifier is to the ideal one depends on how well the OOD samples follow the in-distribution boundary. We find that the method proposed in Lee et al. (2018a) to generate boundary OOD samples is not diverse enough as evidenced by experiments shown in the appendix (Section. D). Therefore we propose a novel approach for boundary OOD sample generation which is described in the next section that results in better boundary OOD samples that cover the in-distribution boundary quite effectively. This evident from our experiments described in the appendix (Section. D).

Lee et al. (2018a) indeed experiment with adding an explicit reject class instead of using a confident-classifier, but the results are found to be worse. But this is because instead of using the boundary OOD samples they use another natural image dataset called "seen OOD" similar to Hendrycks et al. (2019) to train the classifier. However for images, it is difficult to represent the entire OOD space with a small number of samples such methods may not perform that well. Moreover as pointed out in both Lee et al. (2018a) and Hendrycks et al. (2019), as these "seen OOD" samples aren't diverse, when used to train a reject classifier they can overfit to these training OOD samples. However we use boundary OOD samples that can guide the decision boundary of the classifier to be bounded around the in-distribution regions as depicted in Figure. 1b

Note that both the reject-classifier and the confident-classifier use boundary OOD samples for training. The confident-classifier tries to equalize $f_k$'s for boundary OOD samples (i.e., maximize the en-

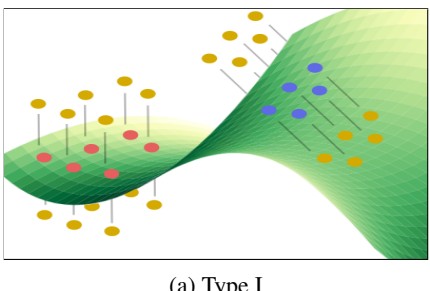
(a) Type I

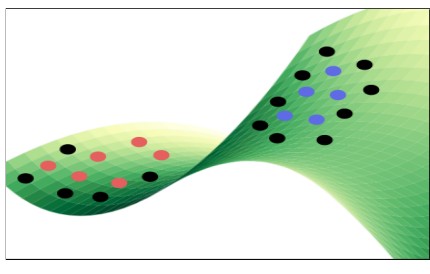
(b) Type II

Figure 3: Categories of OOD samples that we generate: (a) Type I (yellow), which includes samples that are close to the data but outside the in-distribution sub-manifolds, and (b) Type II (black), which includes samples that lie on the in-distribution sub-manifolds and trace the in-distribution boundary; in-distribution clusters are represented through blue and red points.

tropy of output predictions) and expect this to persist over OOD samples far from the in-distribution as well (i.e., have multiple $k^*$'s). The reject-classifier on the other hand maximizes the prediction confidence of $K + 1^{th}$ for boundary OOD samples and expects it to persists over OOD samples far from the in-distribution (i.e., have a single $k^*$ at $k = K + 1$). confidence $f_k$. In the unbounded case, for a confident-classifier, while it is proven that one can almost always find arbitrarily high confidence regions far from the in-distribution, for a reject classifier we can still expect those OOD samples to be classified as belonging to the $K + 1^{th}$ class. In the bounded case too as shown previously, one can obtain decreasingly low entropy regions far from the in-distribution for the confident classifier whereas for the reject-classifier it is similar to the unbounded case. As evident from the experimental results in Figure. 5 we can indeed find OOD samples with low entropy for the confident-classifiers without stretching to infinity, whereas for the reject-classifier, all those OOD samples far away from the in-distribution are correctly classified as OOD. The results in Table. 1 further reinforces the superiority of the reject-classifier over the confident classifier.

## 5 OUT-OF-DISTRIBUTION SAMPLE GENERATION

The proposed approach leverages the following generic assumptions (Cayton (2005), Narayanan & Mitter (2010), Rifai et al. (2011)) that hold true for a wide range of problems, primarily for image data, which is the data used to validate our approach.

**The manifold hypothesis** states that the higher dimensional real-world data in the input space is likely concentrated on a much lower-dimensional sub-manifold.

**The multi-class manifold hypothesis** states that, if data contains multiple classes, different classes correspond to disjoint sub-manifolds separated by low-density regions in the input space.

To fully cover the "boundary" of in-distribution, we identify two categories of OOD samples that are to be generated. As shown in Figure 3, **Type I)** are the OOD samples that are close but outside the in-distribution sub-manifolds; **Type II)** are the OOD samples that are on the sub-manifolds but close to the "boundary" of the in-distribution.

### 5.1 OOD SAMPLES OUTSIDE THE DATA MANIFOLD

These samples are obtained by adding small perturbations to in-distribution samples that are concentrated on the manifold. These perturbations should be added in directions such that the resulting samples should fall outside the manifold. The directions locally normal to the data-supporting manifold can be thought of as the directions that are less likely to contain in-distribution samples and the tangent directions as the more likely ones. Therefore we add perturbations in the normal directions to get OOD samples.

Deep generative models such as VAEs (Kingma & Welling, 2013) and GANs (Goodfellow et al., 2014b) can model the data manifold of observations $x \in X$ through corresponding latent variables $z \in Z$ via a mapping function $g : Z \rightarrow X$ as $x = g(z)$. With a choice of reasonably lower

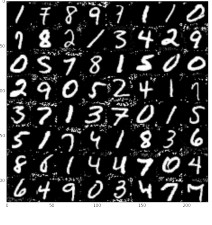 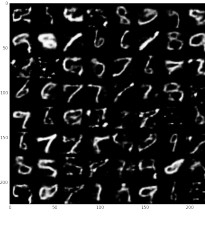 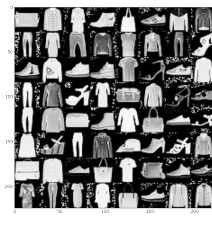 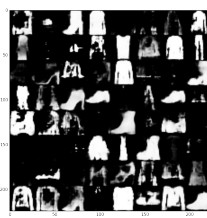

(a) MNIST (Type I)        (b) MNIST (Type II)        (c) F-MNIST (Type I)        (d) F-MNIST (Type II)

Figure 4: Generated OOD samples using the proposed method; Type I OOD samples typically modify the background pixels (normal components have the least variance), while Type II OOD samples modify the object pixels.

dimensional $z$ and a flexible generative function $g$, the model can efficiently represent the true data manifold. Following the **multi-class manifold hypothesis**, we use a conditional generative model that is conditioned over the class labels. For our experiments, we use a conditional variational auto-encoder (CVAE).

Let $h : X \to Z$ and $g : Z \to \hat{X}$ denote the encoder and decoder functions of CVAE respectively. The tangent space of the manifold at a point $x \in X$ is given by the column space of the Jacobian[4]

$$\boldsymbol{J}(\boldsymbol{x}) = \frac{\partial g(\boldsymbol{z})}{\partial \boldsymbol{z}}\bigg|_{\boldsymbol{z} = h(\boldsymbol{x})} \tag{2}$$

Let $\boldsymbol{N}(\boldsymbol{x})$ denote the null-space of $\boldsymbol{J}^T(\boldsymbol{x})$ (left null space of $\boldsymbol{J}(\boldsymbol{x})$). Then the basis vectors of $\boldsymbol{N}(\boldsymbol{x})$ span the normal bundle of the manifold at $\boldsymbol{x}$. Let $\boldsymbol{v}(\boldsymbol{x}) \sim \boldsymbol{N}(\boldsymbol{x})$ be a randomly sampled unit vector from $\boldsymbol{N}(\boldsymbol{x})$, then the perturbed sample is given by,

$$\tilde{\boldsymbol{x}} = \boldsymbol{x} + \beta \boldsymbol{v}(\boldsymbol{x}) \tag{3}$$

where $\beta \in \mathbb{R}$ is a hyper-parameter that controls how far the perturbed sample is from the in-distribution point. In our experiments, we use a stochastic $\beta$ that is uniformly sampled from in the range $[0.1, 1.0]$. As discussed before, for better OOD detection, the boundary samples generated should be diverse; because the proposed approach generates OOD samples by randomly perturbing every in-distribution training sample, the diversity of the generated samples is ensured. This is visually apparent from the experimental results on a 3D-dataset shown in section D of appendix. Figure 4 illustrates the perturbed samples for MNIST and Fashion MNIST datasets. One can observe that the perturbations added mostly modify the background pixels than the object pixels. This is because the normal directions to the manifold mostly represent least variance components of the image.

## 5.2    OOD SAMPLES ON THE DATA MANIFOLD

These are the samples that are in the low-density regions of the input space but close to the in-distribution boundaries on the manifold.

For a variational auto-encoder, the aggregate posterior $q(\boldsymbol{z})$ (Makhzani et al., 2015) is given by,

$$q(\boldsymbol{z}) = \int_{\boldsymbol{x}} q(\boldsymbol{z}|\boldsymbol{x}) p_{in}(\boldsymbol{x}) d\boldsymbol{z} \tag{4}$$

where $p_{in}(\boldsymbol{x})$ is the probability density function of in-distribution and $q(\boldsymbol{z}/\boldsymbol{x})$ is the approximate posterior. Assuming a smooth decoder, the high-density regions in the aggregate posterior can be thought of as corresponding to densely populated regions in the input space, and the input space density would gradually decrease as we sample away from the high-density regions in the aggregate posterior. Therefore the in-distribution boundary on the manifold can be approximated by regions at a distance away from the high-density areas where the density dips below a certain threshold. For our experiments, we approximate $q(\boldsymbol{z})$ with a uni-modal Gaussian distribution whose mean

---

[4]While $\boldsymbol{z}$ is stochastic, we just use its mean estimate for generating OOD samples outside the manifold.

$\hat{\mu}$ and co-variance $\hat{\Sigma}$ are estimated using the encoder mappings of in-distribution samples. We use Mahalanobis distance as a criterion to determine the distance from the mean to sample and generate the required OOD samples. Let $r$ be the Mahalanobis distance from the mean of $q(\boldsymbol{z})$ that encompasses 95% of the training data. The OOD samples are generated by decoding the uniformly sampled samples from the latent space over the surface of a hyper-ellipsoid (Rubinstein, 1982) defined by 5, where $\hat{\mu}_z$ and $\hat{\Sigma}_z$ are the mean and co-variance estimates of $q(z)$, respectively.

$$(\boldsymbol{z} - \hat{\mu}_z)^T \hat{\Sigma}_z^{-1} (\boldsymbol{z} - \hat{\mu}_z) = r^2 \tag{5}$$

It is fair to assume a uni-modal Gaussian distribution for $q(\boldsymbol{z})$ as we fit a Gaussian per class. Moreover, a substantial gain in the ODD detection results when the classifier is trained with these samples can also be taken as evidence pointing towards the validity of such an assumption.

The generated OOD samples described in 5.1 and 5.2 are then used to train an $n+1$ class softmax classifier, where the $n + 1^{th}$ class represents the OOD class.

## 6 EXPERIMENTS

Experiments[5] are divided into 2 sections; the first section explains the toy experiments on a low-dimensional dataset to support our theoretical analysis of the confident-classifier, the second section gives details of OOD detection experiments on MNIST and Fashion MNIST ((Xiao et al., 2017)) using the proposed method.

### 6.1 LIMITATIONS OF CONFIDENT-CLASSIFIERS

In these experiments, the input space is $\mathbb{R}^2$ and the in-distribution consists of 2-classes. The samples for each of these classes are generated by sampling from 2 Gaussians with identity co-variances and means (-10, 0) and (10, 0) respectively, on the Cartesian coordinates. Anything outside 3 standard deviations (Mahalanobis distance) from the in-distribution means is considered OOD. The architecture of the neural network used is similar to the one used in Lee et al. (2018a), which is a ReLU-classifier with 2 fully-connected hidden layers with 500 neurons each.

Following the case in Lee et al. (2018a), for training, OOD samples are generated close to the in-distribution as shown in Figure 5a. For testing, OOD samples are uniformly sampled from a 2D box $[-50, 50]^2$ excluding the in-distribution regions.

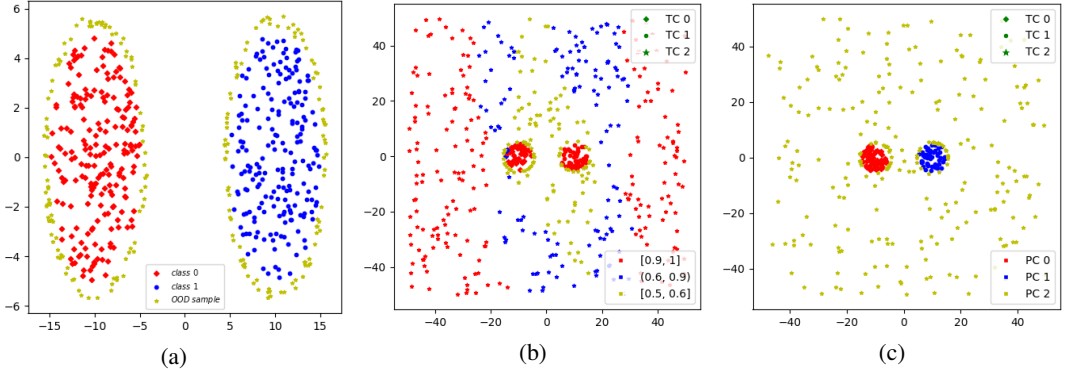

Figure 5: Plots for boundary OOD samples experiments. (a) Training data in 2D. (b) Maximum prediction output on test data for a confident-classifier. (c) Classification output of a classifier with a "reject" class on test data (TC = true class, PC = predicted class).

From Figure 5b, we observe that the ReLU-classifier trained to optimize confidence loss results in highly confident predictions for many OOD samples far from the in-distribution data. This renders the classifier ineffective at classifying the in and out of distribution samples based on the maximum prediction score (confidence) or the entropy of the output. However, from Figure 5c, for a classifier

---

[5]Code: https://github.com/iclr2020-ai/ICLR2020

trained with explicit reject class, the test OOD samples are indeed classified as OOD. This supports the aforementioned intuitions in 4.

Note that these are not the results specific to a certain architecture of the neural network. Experiments with different hyper-parameters such as the number of hidden neurons, changing input dimensions, using sigmoid activation functions instead of ReLU lead to similar results. We remark however that for sigmoid networks, the results were not as extreme (in terms of the number of OOD samples with high-confidence) as for ReLU networks. This is understandable because sigmoid activation outputs will not produce arbitrarily large values, unlike the ReLU counterparts.

## 6.2 MNIST AND FASHION MNIST EXPERIMENTS

We validated our approach on MNIST and Fashion MNIST as in-distribution datasets and several other OOD datasets. For all MNIST as in-distribution experiments, we use a CVAE with a latent dimension of 8, and for Fashion MNIST, the latent dimension is set to 10. We compare our approach against the recent classifier-based OOD detectors such as confident-classifier, ODIN and Mahalanobis distance-based approach without feature ensemble (MD)[6]. The architecture for both CVAE and the classifier used are shown in the appendix. Both the networks are trained till convergence.

### 6.2.1 OOD DATASETS

MNIST is used as an OOD dataset for Fashion MNIST as in-distribution, and vice-versa. For MNIST 0-4 experiment, we use images in class 0 through 4 as in-distribution and class 5 through 9 as OOD. We use both character datasets and noise generated images as OOD datasets. The character datasets are Omniglot (Lake et al., 2015), EMNIST-letters (Cohen et al., 2017) and NotMNIST (Bulatov, 2011). The noise generated images are described below.

**Gaussian noise** includes gray-scale images, where each pixel is sampled from an independent normal distribution with 0.5 mean and unit-variance.

**Uniform noise** includes gray-scale images where each pixel is sampled from an independent uniform distribution in the range $[0, 1]$.

**Sphere OOD** contains images sampled from the surface of a 784 dimensional hyper-sphere centered at the origin with a radius equal to the maximum Euclidean distance of in-distribution samples from the origin and reshaped to $28 \times 28$. This is used to show the effectiveness of our approach not only on the datasets that are restricted to a finite range such as $[0, 1]^d$ for images in $[0, 1]^d$ but also for a general case of $\mathbb{R}^d$.

### 6.2.2 EVALUATION METRICS FOR OOD DETECTION

We experimented with two different metrics as OOD score to determine if the given input sample is in or out of distribution. **OOD class probability** is the $n + 1^{th}$ class prediction probability. **In-distribution max probability** is the maximum prediction probabilities of the in-distribution classes. A higher (lower) OOD class probability (in-distribution max probability) indicates a higher probability of a sample being OOD. Except for MNIST 0-4 experiments, we find that the former metric gives the best results. We report only the best score in Table 1. We use the area under the ROC curve (AUROC↑), the area under the precision-recall curve (AUPR↑), the false positive rate at 95% true positive rate (FPR95↓) and the detection error as the metrics for evaluation. These metrics are commonly used for evaluating OOD detection methods (Lee et al., 2018a; Hendrycks et al., 2019). The details of which are in the appendix.

### 6.2.3 DETECTION RESULTS

Table 1 compares our approach with other approaches for experiments on MNIST and Fashion MNIST as in-distribution datasets. Since the classifier is trained with OOD samples, there is a possibility of reduction in the classification accuracy of in-distribution classes in comparison to training without OOD class. We therefore report classification accuracy of a classifier trained with and without OOD samples. We find that there is no significant change in accuracy. Training our method

---

[6]Results for 3 other approaches are in the appendix

Table 1: OOD detection results

| ID Model (acc before OOD/ acc after OOD) | OOD | FPR at 95% TPR ↓ | Detection Error↓ | AUROC↑ | AUPR Out↑ | AUPR In↑ |
|---|---|---|---|---|---|---|
| | | Ours/Confident-Classifier/ODIN/MD | | | | |
| MNIST (99.0/98.9) | F-MNIST | **0.0**/7.9/0.4/94.2 | **0.2**/5.6/1.8/11.9 | **100.0**/98.5/99.8/86.6 | **100.0**/98.8/99.8/92.0 | **100.0**/98.4/99.8/74.0 |
| | EMNIST-letters | **1.6**/31.0/25.7/31.2 | **3.0**/13.2/11.7/13.6 | **99.6**/93.0/94.4/93.2 | **99.6**/93.0/94.3/92.7 | **99.6**/92.4/94.1/93.2 |
| | NotMNIST | **0.0**/26.5/11.3/34.8 | **0.0**/12.3/6.9/16.3 | **100.0**/94.0/97.8/91.7 | **100.0**/93.9/**100.0**/91.7 | **100.0**/93.8/97.7/92.3 |
| | Omniglot | **0.0/0.0/0.0**/98.5 | **0.0**/1.0/0.2/46.9 | **100.0/100.0/100.0**/19.8 | **100.0/100.0/100.0**/40.8 | **100.0/100.0/100.0**/35.0 |
| | Gaussian-Noise | **0.0/0.0/0.0**/99.9 | **0.0/0.0/0.0**/24.6 | **100.0/100.0/100.0**/50.9 | **100.0/100.0/100.0**/71.8 | **100.0/100.0/100.0**/35.1 |
| | Uniform-Noise | **0.0/0.0/0.0**/82.6 | **0.0/0.0/0.0**/26.4 | **100.0/100.0/100.0**/65.0 | **100.0/100.0/100.0**/76.0 | **100.0/100.0/100.0**/63.9 |
| | Sphere-OOD | **0.0**/21.6/0.0/80.4 | **0.1**/6.6/1.4/14.9 | **100.0**/96.8/99.8/87.6 | **100.0**/97.8/99.9/91.7 | **100.0**/95.2/99.8/79.9 |
| F-MNIST (91.9/91.2) | MNIST | 4.1/87.4/70.2/**2.4** | 4.2/36.3/28.9/**3.6** | 98.7/67.0/76.7/**99.5** | 98.2/65.2/73.2/**99.5** | **100.0**/64.8/77.3/99.4 |
| | EMNIST-letters | **6.4**/87.3/83.5/10.1 | **5.4**/41.8/13.6/7.3 | 97.9/61.1/66.6/**98.1** | 96.8/60.0/62.0/**98.3** | **98.5**/61.6/66.6/98.1 |
| | NotMNIST | **0.8**/78.9/80.2/7.2 | **1.2**/32.2/33.9/5.8 | **99.7**/73.7/69.3/97.8 | **99.5**/73.0/63.0/97.4 | **99.8**/72.4/70.5/98.2 |
| | Omniglot | **0.0**/59.8/9.6/58.4 | **0.9**/22.1/7.1/26.8 | 99.8/85.6/97.9/83.2 | **99.9**/85.8/97.6/84.9 | **99.6**/85.1/98.2/83.4 |
| | Gaussian-Noise | **0.0**/32.2/4.5/99.7 | **0.2**/9.6/3.8/19.9 | 99.8/95.8/98.0/80.0 | **99.9**/96.7/96.7/87.0 | **99.5**/94.7/95.6/66.3 |
| | Uniform-Noise | **0.2**/71.0/99.4/1.7 | **1.3**/16.4/24.7/3.3 | **99.8**/88.6/74.7/98.9 | 99.8/91.8/82.9/**99.2** | **99.8**/82.9/61.6/97.9 |
| | Sphere-OOD | 0.6/99.3/100.0/**0.0** | 0.8/50.0/50.0/**0.0** | 99.7/29.6/0.25/**100.0** | 99.4/39.1/30.7/**100.0** | 99.8/37.4/30.7/**100.0** |
| MNIST0-4 (99.8/99.6) | MNIST5-9 | **17.2**/21.9/20.4/50.0 | **10.0**/12.0/11.5/14.4 | **95.1**/92.9/93.4/92.3 | **94.0**/92.1/91.3/93.8 | **94.9**/93.6/94.2/90.1 |
| | F-MNIST | **0.2**/1.7/2.0/41.4 | **1.6**/3.1/3.4/15.1 | **99.8**/99.4/99.4/92.5 | **99.8**/99.5/99.4/93.3 | **99.7**/99.3/99.3/91.9 |
| | EMNIST-letters | **2.7**/22.1/26.4/12.9 | **3.8**/12.4/13.9/7.6 | **99.2**/92.9/92.3/96.9 | **99.3**/92.0/90.4/96.6 | **99.1**/93.6/93.2/97.1 |
| | NotMNIST | **0.0**/10.9/28.0/2.8 | **0.1**/7.7/13.3/3.1 | **100.0**/97.5/93.5/99.3 | **100.0**/97.5/92.7/99.2 | **100.0**/97.6/93.7/99.4 |
| | Omniglot | **0.0/0.0**/2.3/**0.0** | **0.0/0.1**/3.6/0.4 | **100.0/100.0**/99.1/**100.0** | **100.0/100.0**/99.3/**100.0** | **100.0/100.0**/98.8/**100.0** |
| | Gaussian-Noise | **0.0/0.0/0.0**/0.2 | **0.0/0.0**/0.1/2.4 | **100.0/100.0/100.0**/97.5 | **100.0/100.0/100.0**/98.6 | **100.0/100.0**/99.7/92.2 |
| | Uniform-Noise | **0.0/0.0/0.0**/25.9 | **0.0/0.0**/0.4/5.1 | **100.0/100.0**/99.9/95.9 | **100.0/100.0**/99.9/97.6 | **100.0/100.0**/99.6/89.4 |
| | Sphere-OOD | **0.0**/7.1/0.2/22.7 | **0.1**/5.5/2.0/6.9 | **100.0**/98.2/99.6/96.5 | **100.0**/98.6/99.7/97.6 | **100.0**/97.4/99.3/93.8 |
| F-MNIST0-4 (94.2/94.8) | F-MNIST5-9 | **19.7**/55.8/29.2/75.8 | **12.3**/17.1/14.6/26.4 | **92.5**/89.5/92.1/79.5 | 88.7/90.2/**91.3**/79.8 | **94.3**/87.1/92.8/77.7 |
| | MNIST | **1.8**/67.3/53.5/2.0 | **2.3**/23.6/21.1/3.4 | **99.5**/83.5/86.4/99.0 | **99.4**/84.2/86.1/99.3 | **99.6**/81.7/85.7/98.5 |
| | EMNIST-letters | **1.2**/71.6/48.4/14.1 | **2.4**/24.2/20.3/7.6 | **99.6**/82.6/87.7/98.0 | **99.6**/83.8/87.7/98.0 | **99.7**/79.8/87.9/96.9 |
| | NotMNIST | **0.2**/76.0/57.7/11.0 | **1.2**/26.8/23.6/8.0 | **99.9**/79.9/84.1/97.0 | **99.8**/81.3/83.8/96.8 | **99.9**/77.1/83.9/97.2 |
| | Omniglot | **1.0**/62.3/15.5/11.1 | **2.5**/18.3/9.1/7.1 | **99.5**/88.6/96.5/97.5 | **99.6**/90.6/96.0/97.9 | **99.3**/85.8/96.7/95.7 |
| | Gaussian-Noise | **0.0**/0.3/**0.0**/99.3 | **0.4**/2.0/**0.4**/41.7 | **100.0**/99.7/**100.0**/53.4 | **100.0**/99.8/**100.0**/62.6 | **100.0**/99.7/**100.0**/47.9 |
| | Uniform-Noise | **0.0**/9.8/1.3/36.3 | **0.3**/5.4/3.0/8.5 | **100.0**/98.1/99.2/95.0 | **100.0**/98.6/99.4/96.6 | **100.0**/97.5/98.9/90.1 |
| | Sphere-OOD | **0.0**/89.6/95.5/**0.0** | **0.0**/38.3/41.6/**0.0** | **100.0**/65.8/59.8/**100.0** | **100.0**/67.7/62.4/**100.0** | **100.0**/61.9/55.0/**100.0** |

requires tuning hyper-parameter such as $\beta$ from Eq. 4, OOD class weight, and learning rate. The hyper-parameters were chosen based on the in-distribution classification accuracy and the AUROC of the validation generated OOD samples and the random noise datasets. For all our experiments we use a stochastic $\beta$ uniformly sampled in the range [0.1, 1], OOD class weight is set to 0.1, while the weights for the rest of the classes is set to 1.0, and Adadelta (Zeiler, 2012) is the optimizer used with learning rates of 0.1 and 0.01 for Fashion-MNIST and MNIST experiments, respectively. We do not tune the hyper parameters per OOD dataset unlike ODIN and Mahalanobis distance-based approaches, where the perturbation magnitude is tuned per OOD dataset. Even without this advantage, our method still performs better than these baselines for most of the OOD datasets.

We would like to remark that our approach gives good OOD detection results consistently on all the OOD datasets used unlike the baselines compared. This indicates that our approach is robust to change in OOD datasets.

## 7 CONCLUSION

We have shown in the paper that the confident-classifier almost always has OOD samples that produce high confidence outputs (in the contexts described earlier). We provided empirical evidence that favor using an explicit "reject" class instead. However, the ODD detection capabilities of a reject-classifier depend on the extent to which the generated OOD samples follow the low-density boundaries of in-distribution. We also propose a novel algorithm for generating "effective" OOD samples for training an $n + 1$-class classifier for OOD detection and the results for most of the experiments on gray-scale datasets are consistently better for our approach in comparisons to other methods compared. For future research, we would like to investigate its effectiveness on non-gray-scale datasets such as CIFAR and TinyImageNet. However we would like to point out that the nullspace calculation for colored images is computationally quite expensive, hence requires a larger compute (refer appendix E).

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

# A    APPENDIX

# B    RELATED WORK

There have been many approaches in the literature proposed to address the problem of OOD detection in the context of image data. Most of the successful ones are either generative ((Pidhorskyi et al., 2018; Wang et al., 2017; Ren et al., 2019)) or classifier-based approaches (Hendrycks & Gimpel, 2016; Hendrycks et al., 2019; DeVries & Taylor, 2018; Liang et al., 2018; Lee et al., 2018b). Generative approaches either explicitly or implicitly estimate the input density or use reconstruction error as a criterion to decide if input belongs to OOD. Classifier-based approaches, on the other hand, incorporate OOD detection as a part of the classifier network. The approach proposed in this paper belongs to the latter category. Therefore we limit our related work discussion to only classifier-based approaches.

Typical discriminatively trained classifiers that model the conditional probability $P(y|x)$ without any additional constraints, by definition can make reliable classification decisions only on in-distribution data. For out-of-distribution data, the classifier output is arbitrary. Moreover, any meta information from the output of the classifier or the features learned are also conditioned on the data belonging to in-distribution. Therefore this information in-principle cannot be used to ascertain if

the input is in or out of distribution. However, most of the recent approaches ((Hendrycks & Gimpel, 2016; Lee et al., 2018b; Liang et al., 2018; DeVries & Taylor, 2018))in the literature follow this approach.

Hendrycks & Gimpel (2016) propose a baseline approach to detect OOD inputs, called max-softmax by thresholding the maximum softmax output of a pre-trained classifier. Liang et al. (2018) improve upon this using temperature scaling (ODIN, (Guo et al., 2017)) and adding input perturbations. The assumptions is that these changes result in larger separation between in and out of distribution data in terms of their output predictions. Lee et al. (2018b) propose an approach based on the assumption that the class-conditional features of a softmax classifier follow a Gaussian distribution. Therefore, Mahalanobis distance (MD) from the mean of the Gaussian is used as a score to detect OOD. This is then combined with input perturbations similar to ODIN to enhance the OOD detection results. This method obtains state-of-the-art results on most of the baseline datasets used in OOD detection literature. Despite good results, the method can be seen as OOD detection on feature space rather than pixel space not conforming to the usual definition of OOD (By definition, the in-distribution, $p_{in}(x)$ is defined for $x \in \mathbb{X}$ in pixel space, and hence OOD is also defined in the same space). Hence the effectiveness of the method highly depends on the features learned by the classifier, and also there is no guarantee that the optimization algorithm forces the features to follow a Gaussian distribution. Hendrycks et al. (2019) propose to train a classifier with a confidence loss where OOD data is sampled from a large natural dataset. Hein et al. (2019) also follow a similar approach using a confidence loss and uniformly generated random OOD samples from the input space. In addition, they not only minimize the confidence at the generated OOD samples, but also in the neighbourhood of those samples. However, because both these approaches use the confidence-loss, they suffer from the problems explained in this paper. Moreover, such approaches are only feasible for input spaces where it is possible to represent the support of OOD with finite samples (assuming uniform distribution over OOD space). This is not possible when the input space is $\mathbb{R}^d$, whereas the method proposed in this paper is still applicable.

Geifman et al. (2018) propose to use Bayesian prediction uncertainties given by MC-Dropout (Gal & Ghahramani, 2016) for OOD detection. However, on the theoretical front, the Bayesian uncertainty measure only characterizes the uncertainty in in-distribution. Therefore in principle should not be applied to OOD detection.

## C   TOY EXPERIMENT ON GENERAL OOD SAMPLES

In this case, both train and test OOD samples are uniformly sampled from a 2D box $[-50, 50]^2$ excluding the in-distribution regions. From Figure 6, we observe that both confidence loss and reject class based classifiers are able to distinguish in and out of distribution samples effectively. Therefore, there is no clear winner between the two. However as mentioned previously, such approaches are only feasible for input spaces where (approximately) representing the entire OOD region with a finite number of samples is possible. This is definitely not possible for example when the input space is $\mathbb{R}^d$.

## D   GENERATING OOD SAMPLES USING A GAN VS OUR APPROACH

Lee et al. (2018a) propose to generate OOD samples in the low-density regions of in-distribution by optimizing a joint GAN-classifier loss, (1). With a toy experiment, they show that the generator indeed produces such samples and also these samples follow the "boundary" of the in-distribution data. However, in the experiment, they use a pre-trained classifier. The classifier is pre-trained to optimize the confidence loss on in-distribution and OOD samples sampled close to the in-distribution. Therefore the classifier already has the knowledge of those OOD samples. When GAN is then trained following the objective in (1), GAN likely generates those OOD samples close to the in-distribution. But it is evident that this setting is not realistic as one cannot have a fully informative prior knowledge of those OOD samples if our objective is to generate them.

Therefore, we experiment by directly optimizing (1) where the classifier is not pre-trained. The in-distribution data for the experiment is obtained by sampling over the surface of a unit sphere from its diagonally opposite quadrants to form 2 classes respectively as shown in Figure 7. We find that (with much hyper-parameter tuning), even though GAN ends up producing OOD samples close to

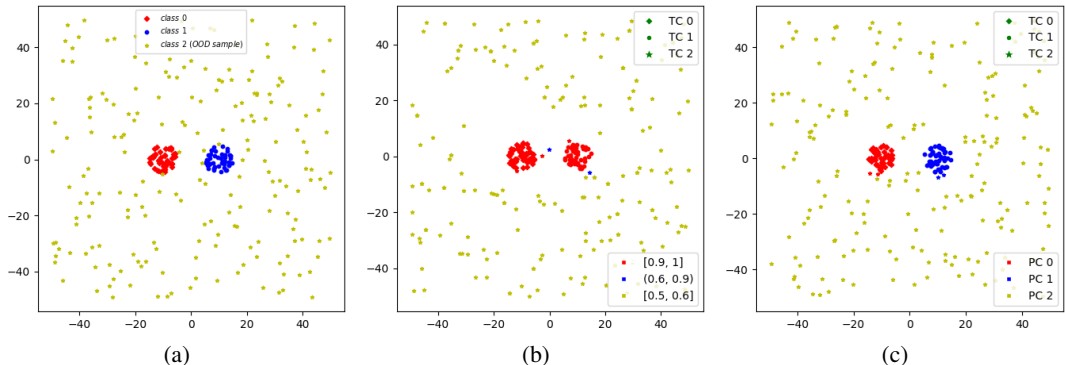

Figure 6: Plots for general OOD samples experiments. (a) Training data in 2D. (b) Maximum prediction output on test data for a confident-classifier. (c) Classification output of a classifier with a "reject" class on test data.

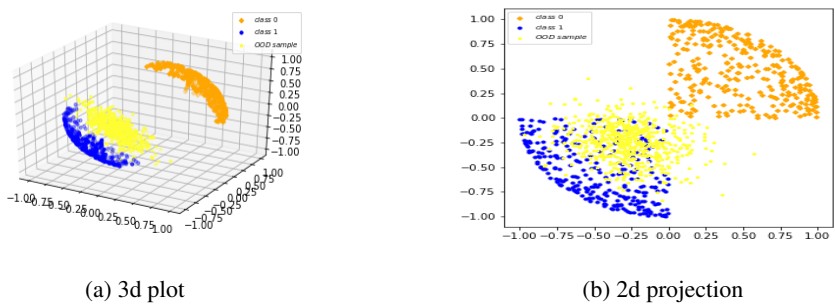

(a) 3d plot                                    (b) 2d projection

Figure 7: Generated OOD samples using a joint training of a GAN and a confident-classifier. We observe that the generated OOD samples don't cover the entire in-distribution boundary.

the in-distribution, it does an unsatisfactory job at producing samples that could follow the entire in-distribution boundary. Moreover, there is less diversity in the generated samples which make them ineffective at improving the classifier performance in OOD detection. Our intuition is that the loss (1(b)+1(c)) that forces the generator of the GAN to generate samples in the high entropy regions of the classifier doesn't necessarily enforce it to produce samples that follow the entire in-distribution boundary. The inability of GANs to generate such samples for a simple 3D dataset indicates that it would be even more difficult in higher dimensions.

In comparison to the GAN based boundary OOD generation, our approach as visually apparent from Figure 8 produces samples that cover the in-distribution boundary quite effectively. While it is difficult to visualize how well the off-manifold OOD samples cover the boundary, one can imagine them having a good coverage on the off-manifold boundary as they are obtained by perturbing each training sample in the direction given by the null-spaces. Hence the diversity of the OOD samples is ensured. For on-manifold boundary OOD samples, as evident from Figure 8c, it forms a closed boundary around the in-distribution points.

# E   DISCUSSION ON SAMPLE GENERATION COMPLEXITY

For generating OOD samples outside the manifold, we randomly sample from the left-null-space of the Jacobian as described earlier. But the complexity of this step depends on the number of basis vectors in the null-space and its dimensions. For the MNIST case, with the input dimensions $28 \times 28$ and latent dimension of 8, there are 776 basis vectors in the left-nullspace, each of dimension 784 (i.e., $28 \times 28$). For colored images such as CIFAR and TinyImagenet, the number of basis vectors are almost 3 times of that for gray-scaled images. To cover the in-distribution boundary effectively in all directions, many OOD samples for each in-distribution training sample are to be generated by taking

random linear combinations of the basis vectors, which is quite expensive. This gives a quantitative measure of effective OOD sample complexity. However, we find that only a few OOD samples are sufficient to guide the decision boundary of the classifier to be bounded around the in-distribution regions evidenced by their OOD detection results.

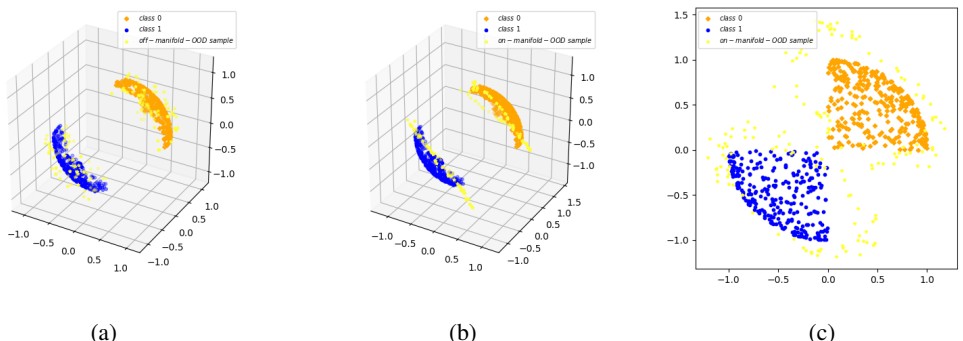

Figure 8: Generated boundary OOD samples using our approach. (a) 3d plot of in-distribution data with out-of-manifold boundary OOD samples. (b) 3d plot of in-distribution data with on-manifold boundary OOD samples. (c) 2d projection of in-distribution data with on-manifold boundary samples to show that they cover the in-distribution boundary on the manifold.

## F    EXPERIMENTAL ARCHITECTURE

The encoder and the decoder parts of the CVAE architecture, and the classifier used are described in Figure 9a, 9b and 9c respectively. The latent dimension ($d$) is chosen per dataset. For MNIST, $d = 8$ and for Fashion MNIST, $d = 10$. The number of features after the convolutions in the encoder is represented by $f$. "cond_x" is the one hot representation of class labels. $k$ in the classifier architecture represents the number of classes in the training data.

## G    METRICS DEFINITIONS

The definitions of metrics used to evaluate OOD detection are as follows.

**FPR at 95% TPR** is the probability of an OOD input being misclassified as in-distribution when 95% of in-distribution samples are correctly classified as in-distribution (i.e, the true positive rate (TPR) is at 95%). True positive rate is calculated as, $TPR = \frac{TP}{TP+FN}$, where TP and FN denote the true positives and false negatives, respectively. The false positive rate (FPR) is computed as $FPR = \frac{FP}{FP+TN}$, where FP and TN denote the false positives and true negatives, respectively.

**Detection error** is the minimum mis-classification probability over all possible thresholds over the OOD score. We assume that the test set contains equal number of in and out of distribution samples.

**AUROC** is the area under the receiver operating characteristic curve, which is a threshold independent metric. ROC curve is a plot of TPR versus FPR. AUROC can be interpreted as the probability that a positive example is assigned a higher detection score than a negative example. For a perfect detector, AUROC is 100%.

**AUPR** is the Area under the Precision-Recall (PR) curve. PR curve is a plot of precision ($TP = (TP + FP)$) versus recall ($TP = (TP + FN)$). The metric AUPR-In and AUPR-Out represent the area under the PR curve depending on if in or out of distribution data are specified as positives, respectively.

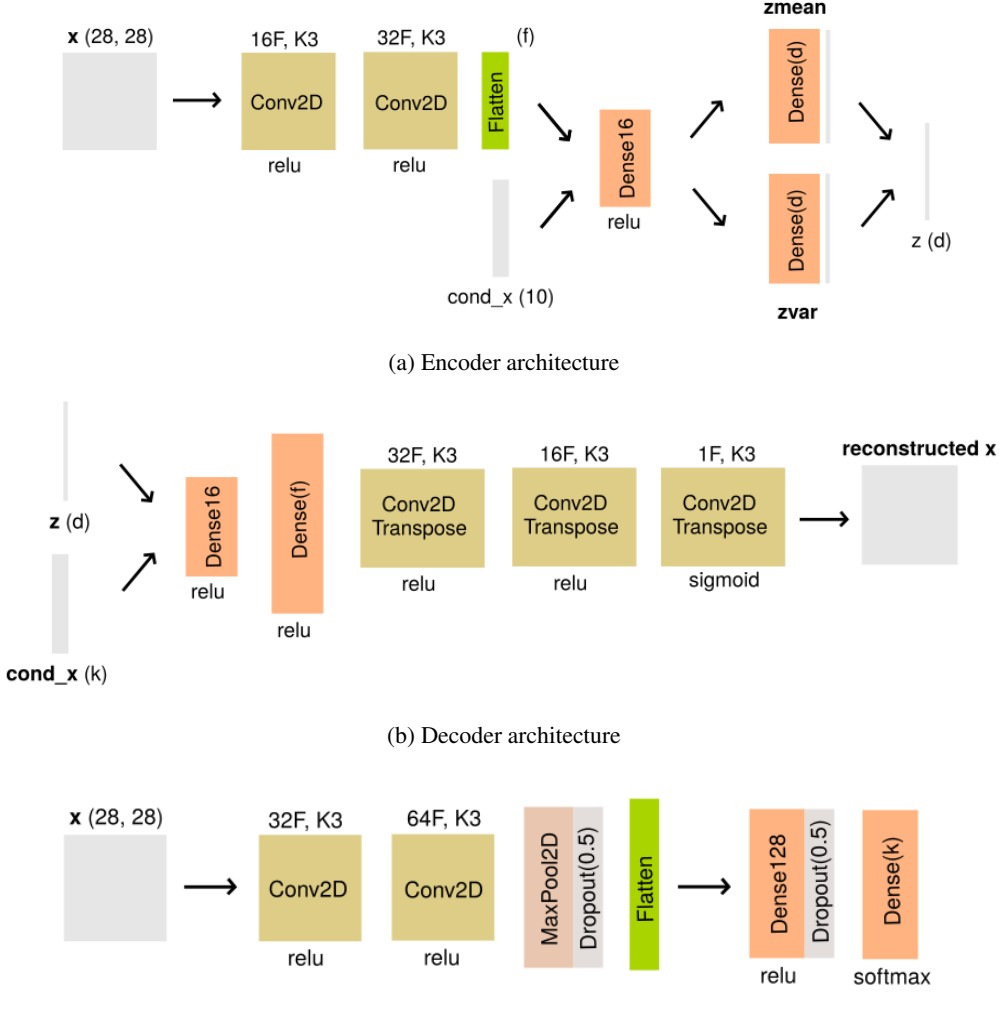

(a) Encoder architecture

(b) Decoder architecture

(c) Classifier architecture

## H    MORE RESULTS

We report the OOD detection results for OOD detection methods based on softmax-score (Hendrycks & Gimpel (2016)), uncertainty of classifier obtained via MC-dropout (Gal & Ghahramani (2016)), and mutual information between predictions and model posterior (Gal et al. (2017)). The results are shown in Table 2.

Table 2: Results

| ID Model | OOD | FPR at 95% TPR ↓ | Detection Error↓ | AUROC↑ | AUPR Out↑ | AUPR In↑ |
|---|---|---|---|---|---|---|
| | | Softmax/MC-Dropout/Mutual-Info | | | | |
| MNIST | F-MNIST | 1.61/2.1/54.3 | 3.3/3.5/14.5 | 99.5/99.4/90.8 | 99.5/99.5/91.5 | 99.5/99.4/86.2 |
| | EMNIST-letters | 28.0/24.5/22.0 | 12.6/11.2/11.1 | 93.6/94.6/95.0 | 93.4/94.4/94.7 | 93.0/94.1/94.7 |
| | NotMNIST | 13.1/12.5/22.1 | 7.1/6.7/9.1 | 97.4/97.6/95.5 | 97.8/97.9/96.0 | 97.0/97.1/94.0 |
| | Omniglot | 0.0/0.0/92.8 | 0.5/0.6/19.5 | 100.0/100.0/84.5 | 100.0/100.0/88.7 | 100.0/100.0/73.9 |
| | Gaussian-Noise | 0.0/0.0/100.0 | 0.0/0.0/49.3 | 100.0/100.0/17.2 | 100.0/100.0/37.7 | 100.0/100.0/34.0 |
| | Uniform-Noise | 0.0/0.0/100.0 | 0.0/0.0/45.2 | 100.0/100.0/45.2 | 100.0/100.0/56.4 | 100.0/100.0/42.9 |
| | Sphere-OOD | 0.8/1.3/7.6 | 2.6/3.1/4.2 | 99.3/99.2/97.3 | 99.5/99.4/99.3 | 99.1/99.0/93.1 |
| F-MNIST | MNIST | 84.5/68.9/19.4 | 34.5/25.0/11.9 | 70.6/82.2/93.5 | 70.9/83.0/91.3 | 68.2/80.3/94.9 |
| | EMNIST-letters | 88.1/77.8/37.6 | 41.1/33.1/21.1 | 62.3/73.4/84.2/ | 62.3/73.2/79.6 | 61.3/72.2/87.9 |
| | NotMNIST | 83.1/67.0/24.2 | 35.2/25.1/14.3 | 68.9/81.7/91.7 | 66.4/80.3/87.6 | 68.6/80.8/93.6 |
| | Omniglot | 39.0/32.5/26.8 | 17.1/14.8/10.4 | 91.4/93.5/95.2 | 91.4/93.7/95.6 | 91.8/95.5/93.1 |
| | Gaussian-Noise | 99.1/98.5/72.2 | 17.5/15.8/9.6 | 80.0/82.1/92.5 | 87.8/89.1/95.3 | 65.5/67.9/83.4 |
| | Uniform-Noise | 96.9/96.5/48.8 | 29.4/24.2/16.8 | 70.1/76.8/90.9 | 78.6/84.0/92.5 | 58.11/64.8/87.9 |
| | Sphere-OOD | 97.2/71.2/1.8 | 50.0/17.0/2.5 | 48.4/88.2/99.6 | 50.6/91.4/99.5 | 47.7/82.4/99.6 |
| MNIST0-4 | MNIST5-9 | 18.2/15.7/15.3 | 10.6/9.7/9.9 | 94.2/94.8/94.5 | 93.0/93.1/92.8 | 94.8/95.4/95.1 |
| | F-MNIST | 3.1/4.4/8.8 | 4.0/4.6/5.9 | 99.0/98.8.97.8 | 99.2/99.0/98.4 | 98.7/98.4/96.6 |
| | EMNIST-LETTERS | 25.3/21.6/21.4 | 12.8/12.2/12.2 | 92.8/93.5/93.4 | 90.6/91.6/91.3 | 93.3/93.9/94.1 |
| | NotMNIST | 21.4/16.2/15.9 | 10.44/9.3/9.4 | 95.5/96.4/96.5 | 95.5/96.4/96.1 | 95.0/95.9/96.4 |
| | Omniglot | 0.2/0.1/0.5 | 1.7/1.9/2.4 | 99.4/99.4/99.2 | 99.6/99.6/99.4 | 98.9/99.0/98.8 |
| | Gaussian-Noise | 0.0/0.0/0.0 | 0.3/0.4/1.0 | 99.7/99.7/98.8 | 99.8.99.8/99.4 | 98.9/98.9/95.8 |
| | Uniform-Noise | 0.0/0.0/0.0 | 0.6/0.9/2.2 | 99.7/99.6/98.3 | 99.8/99.8/99.0 | 99.1/99.0/95.2 |
| | Sphere-OOD | 1.0/1.9/2.5 | 2.8/3.4/3.6 | 99.2/99.0/98.6 | 99.4/99.3/99.1 | 98.6/98.5/97.5 |
| F-MNIST0-4 | F-MNIST5-9 | 73.5/67.1/32.8 | 26.1/23.4/17.3 | 80.1/83.4/89.8 | 80.4/83.3/87.6 | 78.1/82.0/91.2 |
| | MNIST | 44.9/43.9/73.9 | 15.9/16.0/16.9 | 91.0/91.4/87.7 | 91.2/91.3/89.5 | 90.4/90.5/82.3 |
| | EMNIST-letters | 69.8/66.6/43.1 | 26.7/25.4/20.5 | 80.4/82.2/87.4 | 81.1/82.8/86.5 | 79.6/81.5/97.9 |
| | NotMNIST | 71.9/67.0/38.6 | 24.3/20.8/17.0 | 82.8.86.1/90.8 | 84.3/87.7/90.4 | 80.2/83.4/91.0 |
| | Omniglot | 44.8/41.3/29.6 | 15.3/14.0/11.5 | 91.6/93.0/94.7 | 92.1/93.8/94.9 | 91.0/92.2/93.9 |
| | Gaussian-Noise | 0.3/0.5/98.3 | 2.3/2.3/12.2 | 99.8/99.7/87.3 | 99.8/99.8.92.4 | 99.7/99.7.74.2 |
| | Uniform-Noise | 27.7/20.5/8.4 | 8.6/7.9/5.3 | 96.4/97.2/98.0 | 97.2/97.8/98.6 | 95.5/96.5/96.6 |
| | Sphere-OOD | 86.5/67.3/10.7 | 33.7/20.4/7.8 | 71.6/86.2/97.3 | 73.4/88.0/96.7 | 67.3/83.0/97.8 |

