# OpenReview forum: "Improving Confident-Classifiers For Out-of-distribution Detection"
_ICLR.cc/2020/Conference — Reject_

### Official Review · AnonReviewer1 · 2019-10-22
**Official Blind Review #1**

**Rating:** 3

**Review:**

Comments on rebuttal

I don’t think that the authors made a valid argument to address my concerns about theoretical justification and experiments. As I mentioned in the review, the assumption and statements in the paper are not clear to me. Moreover, I think the authors should evaluate their methods on more realistic cases. Because of that, I’d like to keep my score.

====

[Summary]

To detect out-of-distribution (OOD) samples, the authors proposed to add an explicit "reject" class instead of producing a uniform distribution and OOD sample generation method. They showed that the proposed method can perform better than several OOD detectors on MNIST and Fashion-MNIST datasets.

[Detailed comments]

I'd like to recommend a "weak reject" due to the following reasons:

1. Justification is not clear: The authors argue that arbitrarily large confidence values cannot be obtained if there are multiple K*. However, how can we guarantee that there are multiple K* only by introducing the additional class? Could the authors elaborate this more? Also, I'm not sure that the theoretical justifications are really valid because we usually consider bounded input space.

2. Experimental results are not convincing: in the paper, only grayscale datasets, such as MNIST and FMNIST, are considered to evaluate the proposed method and I think it is not enough. I would be appreciated if the authors can provide more evaluations on various datasets (e.g., CIFAR, SVHN, TinyImageNet) and deep architectures (e.g., DenseNet and ResNet) similar to [Hendrycks 19, Liang' 18, Lee' 18].

[Questions]

1. Introducing additional class increases the number of parameters and can suffer from overfitting. Could the authors comment on overfitting issues?

2. Could the authors compare the performance of the proposed method with the ensemble version of MD?

3. Instead of generated OOD samples, could the authors report the performance with explicit OOD samples similar to [Hendrycks' 19]?

[Hendrycks' 19] Hendrycks, D., Mazeika, M. and Dietterich, T.G., Deep anomaly detection with outlier exposure. In ICLR, 2019.

[Lee' 18] Lee, K., Lee, H., Lee, K. and Shin, J., Training confidence-calibrated classifiers for detecting out-of-distribution samples. In ICLR, 2018.

[Liang' 18] Liang, S., Li, Y. and Srikant, R., Enhancing the reliability of out-of-distribution image detection in neural networks In ICLR, 2018.

**Experience Assessment:**

I have published in this field for several years.

**Review Assessment: Checking Correctness Of Derivations And Theory:**

I carefully checked the derivations and theory.

**Review Assessment: Checking Correctness Of Experiments:**

I carefully checked the experiments.

**Review Assessment: Thoroughness In Paper Reading:**

I read the paper thoroughly.

---

> ### Author Response · Authors · 2019-11-11
> **Justifying why reject-classifier option is superior and comment about experiments on colored images.**
>
> First of all thank you very much for the review and, of course, for your time.
> Here are the answers to the questions.
>
> 1. By introducing a reject class, our goal is not to have multiple k*’s. Our goal is to have a single k* such that k* = K+1 so that all the OOD samples are classified as belonging to the reject class. We have added more details in Section 4 to make it clearer.
> As far as the validity of theoretical justifications are concerned, while on page 3, the last paragraph we do mention that our analysis cannot be directly applied to bounded cases to obtain arbitrary confidence values as we cannot make $\alpha_l \rightarrow \infty$, the technique in principle can be applied to obtain high confidence values far from the in-distribution. We have added additional details in section 3 and section 4 that makes it clear that the analysis is still applicable for the case of bounded input space. Even Hein’s et.al., (section 3 last paragraph) on which our analysis is based on also indicate how the analysis holds for the bounded domain evidenced by experiments in Table 2.
>
> 2. We couldn't do these experiments in time as these take larger compute due to nullspace calculation. We have added a section in Appendix E to highlight the same. However, we would like to argue why our approach is still good.
> We focused mainly on following a principled approach for OOD detection unlike many state-of-the-art methods in the literature. We show clearly both with theoretical intuitions and toy experiments on how our approach gives the desired results. Our experiments on gray-scale images are quite comprehensive given that we use a large set of OOD datasets and also report results without tuning hyper-parameter per OOD dataset unlike Mahalanobis and ODIN approaches. Moreover, MNIST0-4 vs MNIST5-9 and F-MNIST0-4 vs F-MNIST5-9 are quite challenging tasks given that the in and out-of-distribution samples in these are quite close to each other. However, our approach does significantly better than the benchmark. Therefore we assume experiments on CIFAR or SVHN may not be necessary to justify our claim about the reject-classifier as other benchmarks that perform so well on these datasets do not perform as well in MNIST and Fashion MNIST experiments. We would like to remark that the OOD detection capability of the reject classifier depends on the quality of boundary OOD samples generated. Therefore if VAE cannot represent these large-scale datasets effectively, the generated OOD samples might also not be a good representation of boundary OOD samples. Therefore the OOD detection accuracy might suffer. In which case, a better boundary OOD sample generation method can improve upon our results. Therefore our work can be judged as a benchmark for OOD detection methods that are based on boundary OOD sample generation.
>
> Questions:
> 1.  Except for adding an extra node at the output, we are not increasing the number of hidden neurons, so overfitting is not an issue.
> 2. Ensemble-MD requires training a logistic regression classifier on top of the original classifier to learn the weights of different layers. For this, they use 10% OOD samples for training (Quoting from MD paper, section 3.2: “Following the similar strategies in [7, 22], we randomly choose 10% of original test samples for training the logistic regression detectors and the remaining test samples are used for evaluation.”), which in our opinion is not practical as in real-world scenarios we don’t have access to OOD samples. Moreover, most of the recent OOD detection papers (Ren ‘19, Abdelzad '19) don’t compare against the ensemble version of MD.
>
> 3. The goal of our approach is to use boundary OOD samples that can guide the decision boundary of the classifier to be bounded around the in-distribution regions as depicted in Figure. 1(b). For images, it is difficult to represent the entire OOD space with a small number of samples, therefore we don’t expect it to give good results. We have added some more information in section 4, paragraph 2 to explain the same. If one is able to use a diverse set of OOD samples that can well represent the OOD space, the reject classifier, and the confident-classifier may perform equally well as evidenced by the toy experiment in Appendix C and Figure. 6.
> Please let us know if anything else is not clear, we would be happy to provide more details.
>
> [Ren ‘19]: Likelihood Ratios for Out-of-Distribution Detection, NeurIPS ‘19
> [Abdelzad ‘19] Detecting Out-of-Distribution Inputs in Deep Neural Networks Using an Early-Layer Output, arXiv preprint, arXiv 1910.10307, 2019

---

### Official Review · AnonReviewer3 · 2019-10-23
**Official Blind Review #3**

**Rating:** 3

**Review:**

** post rebuttal start **

After reading reviews and authors' response, I decided not to change my score.
I recommend to strengthen their theoretical justification or make their method scalable to improve their work.


Detailed comments:

2. "Moreover, their results on some of the gray-scale experiments are significantly worse compared to ours."
-> If you are talking about the comparison in MNIST-variants, please note that experimental results on MNIST cannot be seriously taken unless there is a strong theoretical background; especially, MNIST-variants are too small to talk about the scalability of the method. It is hard to convince readers only with results in MNIST-variants, unless the method has a strong theoretical justification.
However, if your claim is true for general gray-scale images, e.g., preprocessing CIFAR to be in gray scale, then you may add supporting experiments about it.

4. Again, if the method is only applicable to MNIST-variants due to its computational complexity while it has no strong theoretical justification, I can't find benefits from it.

** post rebuttal end **



- Summary:
This paper proposes to improve confident-classifiers for OOD detection by introducing an explicit "reject" class. Although this auxiliary reject class strategy has been explored in the literature and empirically observed that it is not better than the conventional confidence-based detection, the authors provide both theoretical and empirical justification that introducing an auxiliary reject class is indeed more effective.


- Decision and supporting arguments:
Weak reject.

1. Though the analysis is interesting, it is not applicable to both benchmark datasets and real-world cases. Including the benchmark datasets they experimented, the input to the model is in general bounded, e.g., natural images are in RGB format, which is typically normalized to be bounded in [0,1]. Therefore, the polytopes would not be stretched to the infinity in most cases.
On the other hand, note that softmax classifiers produce a high confidence if the input vector and the weight vector of a certain class are in the same direction (of course feature/weight norm also matters, but let's skip it for simplicity). Therefore, if there is an auxiliary reject class, only data in the same direction will be detected as OOD; in other words, OOD is "modeled" to be in the same direction with the weight vector of the auxiliary reject class. However, the conventional confidence-based detection does not model OOD explicitly. Since OOD is widely distributed over the data space by definition, modeling such a wide distribution would be difficult. Thus, the conventional approach makes more sense to me.

2. The experiment is conducted only on MNIST variations, so it is unclear whether their claim is true on large-scale datasets and real-world scenario.
Why don't you provide some experimental results on other datasets commonly used in other OOD detection papers, such as CIFAR, SVHN, TinyImageNet, and so on?


- Comments:
1. In section 4, the authors conjectured the reason why the performance of reject class in Lee et al. (2018a) was worse is that the generated OOD samples do not follow the in-distribution boundaries well. I think Appendix E in the Lee et al.'s paper corresponds to this reasoning, but Lee et al. actually didn't generate OOD samples but simply optimized the confidence loss with a "seen OOD." Lee et al. didn't experiment on MNIST variations but many natural image datasets. So, it is possible that the auxiliary reject class strategy is only effective in MNIST variations. I suggest the authors to do more experiments on larger datasets to avoid this criticism.

**Experience Assessment:**

I have published in this field for several years.

**Review Assessment: Checking Correctness Of Derivations And Theory:**

I assessed the sensibility of the derivations and theory.

**Review Assessment: Checking Correctness Of Experiments:**

I assessed the sensibility of the experiments.

**Review Assessment: Thoroughness In Paper Reading:**

I read the paper at least twice and used my best judgement in assessing the paper.

---

> ### Author Response · Authors · 2019-11-11
> **Clarification on Generated OOD samples being Boundary OOD samples and Experiments on Colored images.**
>
> First of all thank you very much for the review and, of course, for your time. Based on the review summary, we want to elaborate on a few things that might be unclear.
>
> Response to the summary: While the auxiliary reject class option has been explored before, it hasn’t been done by generating OOD samples in the low-density boundary of in-distribution. We clearly mention in our paper at multiple places including the proposed method to generate OOD samples that these samples are the boundary OOD samples. Moreover, we mention in our related work (Appendix B) the following:
> “Hendrycks et al. (2019) propose to train a classifier with a confidence loss where OOD data is sampled from a large natural dataset. Hein et al. (2019) also follow a similar approach using a confidence loss and uniformly generated random OOD samples from the input space. In addition, they not only minimize the confidence at the generated OOD samples but also in the neighborhood of those samples. However, because both these approaches use the confidence-loss, they suffer from the problems explained in this paper. Moreover, such approaches are only feasible for input spaces where it is possible to represent the support of OOD with finite samples (assuming a uniform distribution over OOD space). This is not possible when the input space is $R^d$, whereas the method proposed in this paper is still applicable.”
>
>  Response to supporting arguments:
> 1. While on page 3, the last paragraph we do mention that our analysis cannot be directly applied to bounded cases to obtain arbitrary confidence values as we cannot make $\alpha_l \rightarrow \infty$, the technique in principle can be applied to obtain high confidence values far from the in-distribution. We have added additional details in section 3 and section 4 that makes it clear the analysis is still applicable for the case of bounded input space. Even Hein’s et.al. (section 3 last paragraph), on which our analysis is based on also indicate how the analysis holds for the bounded domain evidenced by their experiments in Table 2.
>
> 2. With respect to OOD being distributed all over the data space, we have already answered this question previously (i.e, we only generated boundary OOD samples and we assume we can model the boundary OOD with few samples).
> For conventional approaches such as Mahalanobis distance-based method or ODIN, we have argued in the related work section (Appendix B), why they cannot be considered as general OOD detection approaches as they work in the discriminative feature space. In fact, we can easily show why these methods don’t work with simple toy experiments in low-dimensional space. Moreover, their results on some of the gray-scale experiments are significantly worse compared to ours.
>
> 4. We couldn't do these experiments in time as these take larger compute due to nullspace calculation. We have added a section in Appendix E to highlight the same. However, we would like to argue why our approach is still good.
> We focused mainly on following a principled approach for OOD detection unlike many state-of-the-art methods in the literature. We show clearly both with theoretical intuitions and toy experiments on how our approach gives the desired results. Our experiments on gray-scale images are quite comprehensive given that we use a large set of OOD datasets and also report results without tuning hyper-parameter per OOD dataset unlike Mahalanobis and ODIN approaches. Moreover, MNIST0-4 vs MNIST5-9 and F-MNIST0-4 vs F-MNIST5-9 are quite challenging tasks given that the in and out-of-distribution samples in these are quite close to each other. However, our approach does significantly better than the benchmark. Therefore we assume experiments on CIFAR or SVHN may not be necessary to justify our claim about the reject-classifier as other benchmarks that perform so well on these datasets do not perform as well in MNIST and Fashion MNIST experiments. We would like to remark that the OOD detection capability of the reject classifier depends on the quality of boundary OOD samples generated. Therefore if VAE cannot represent these large-scale datasets effectively, the generated OOD samples might also not be a good representation of boundary OOD samples. Therefore the OOD detection accuracy might suffer. In which case, a better boundary OOD sample generation method can improve upon our results. Therefore our work can be judged as a benchmark for OOD detection methods that are based on boundary OOD sample generation.
>
> Answers to comments:
>
> 1. Thanks for pointing out the mistake, we have fixed this in the paper. However, it is less likely that we have this problem in our approach as we use boundary OOD samples that can guide the decision boundary of the classifier to be bounded around the in-distribution regions as depicted in Figure. 1(b).
> Please let us know if anything else is not clear, we would be happy to provide more details.

---

### Official Review · AnonReviewer2 · 2019-10-26
**Official Blind Review #2**

**Rating:** 6

**Review:**

The paper proposes an algorithm to generate boundary OOD positive/negative samples to train a classifier for OOD samples. The algorithm is based on the theoretical analysis on why confidence value could be high in unbounded polytopes. CVAE is used as a generative model to get new samples. Experiments are conducted on MNIST and Fashon-MNIST datasets, which are used as OOD and in-distribution, and vice versa. Other datasets are also used for OODs. Comparison are made with Confident-Classifier, ODIN, and Mahalanobis distance-based approach, and the proposed method outperforms the others.
Overall the paper is well-written and well-organized. The proposed method is based on the idea from theoretical analysis, and is reasonable and valid. There are only a couple of things to point out: First, the methods to compare, such as Confident-Classifier and ODIN, are not so strong. Thus, I am not sure whether the performance of the proposed algorithm is dramatically better. Second, I would like to see the sensitivity analysis of the proposed method, because there are several hyper-parameters as mentioned in the paper.
However, I like the method and could be accepted as an ICLR paper.


**Experience Assessment:**

I have published one or two papers in this area.

**Review Assessment: Checking Correctness Of Derivations And Theory:**

I assessed the sensibility of the derivations and theory.

**Review Assessment: Checking Correctness Of Experiments:**

I assessed the sensibility of the experiments.

**Review Assessment: Thoroughness In Paper Reading:**

I read the paper at least twice and used my best judgement in assessing the paper.

---

> ### Author Response · Authors · 2019-11-11
> **Reiterating our contributions and explaining the choices made for benchmark OOD approaches.**
>
> First of all thank you very much for the review and, of course, for your time.  Thanks again for supporting our paper. Going by your review comments it seems you have understood our paper quite well. However, we want to reiterate our major contributions again.
>
> 1. While training with the boundary OOD samples, we propose to use a reject-classifier instead of a confident-classifier and we provide a theoretical argument for the same and also compare our approach against the state-of-the-art classifier-based methods for OOD detection in the literature on MNIST and Fashion MNIST datasets.
>
> 2. We propose a novel method to generate boundary OOD samples that are more diverse and follow the in-distribution boundary better than the one proposed in Lee (confident classifier paper).
>
> Moreover, in our analysis of related work (Appendix B), we have argued why the current state-of-the-art methods in OOD detection such as Mahalanobis-distance based approach and ODIN cannot be considered as general OOD detection approaches as they work in the discriminative feature space. In fact, we can easily show why these methods don’t work with simple toy experiments in low-dimensional space. We, on the other hand, try to obtain the decision boundaries as shown in the Figure. 1b that clearly separates in and out-of-distribution regions.
>
> Now to answer your specific questions,
> 1. The methods to compare, such as Confident-Classifier and ODIN, are not so strong. Thus, I am not sure whether the performance of the proposed algorithm is dramatically better.
>
> While we agree that the confident-classifier results aren't that strong, our approach is however similar to the confident-classifier in that we also generate boundary OOD samples to train the classifier, therefore we compare against it. But Mahalanobis-distance based (state-of-the-art on most datasets) and ODIN are the top classifier-based OOD detection approaches in the literature that recent OOD detection methods ([Ren ‘19], [Abdelzad ‘19]) compare against. Our approach outperforms these methods in most of our experiments without having to fine-tune the hyper-parameters per OOD dataset as these methods do. The detection results are significantly better than Mahalanobis-distance based method, especially for MNIST experiments (compare FPR 95\% values). Moreover, MNIST0-4 vs MNIST5-9 and F-MNIST0-4 vs F-MNIST5-9 are quite challenging tasks given that the in and out-of-distribution samples in these are quite close to each other. However, our approach does significantly better than the benchmark.
>
> 2. I would like to see the sensitivity analysis of the proposed method because there are several hyper-parameters as mentioned in the paper.
>
> The hyper-parameters that we have are $\beta$ from Eq. 2, OOD class weight, and learning rate. The learning rate is a hyper-parameter like in any other classifier training. In our case, it mostly impacts the convergence rate. Since the OOD class has lots more samples than the other in-distribution classes, we used a class-weight to handle class-imbalance. However, an equal mix of classes in each batch of training data performs almost as well as the reported results. β that determines how close to the in-distribution boundary are the OOD samples is very important, as choosing a very large value can make OOD samples to be generated far away from the in-distribution boundaries and a smaller value could make OOD samples fall into the in-distribution regions. But we found that the stochastic $\beta$ chosen in the range [0.1 1.0] works well for all our experiments.
>
> Please let us know if anything else is not clear, we would be happy to provide more details.
>
>
> [Ren ‘19]: Likelihood Ratios for Out-of-Distribution Detection, NeurIPS ‘19
>
> [Abdelzad ‘19] Detecting Out-of-Distribution Inputs in Deep Neural Networks Using an Early-Layer Output, arXiv preprint, arXiv 1910.10307, 2019

---

### Decision · Program_Chairs · 2019-12-19

**Decision:**

Reject

**Comment:**

The paper improves the previous method for detecting out-of-distribution  (OOD) samples.

Some theoretical analysis/motivation is interesting as pointed out by a reviewer. I think the paper is well written in overall and has some potential.

However, as all reviewers pointed out, I think experimental results are quite below the borderline to be accepted (considering the ICLR audience), i.e., the authors should consider non-MNIST-like and more realistic datasets. This indicates the limitation on the scalability of the proposed method.

Hence, I recommend rejection.